# Effect of Quercetin and Fingolimod, Alone or in Combination, on the Sphingolipid Metabolism in HepG2 Cells

**DOI:** 10.3390/ijms232213916

**Published:** 2022-11-11

**Authors:** Albena Momchilova, Georgi Nikolaev, Stefan Pankov, Evgenia Vassileva, Nikolai Krastev, Bozhil Robev, Dimo Krastev, Adriana Pinkas, Roumen Pankov

**Affiliations:** 1Institute of Biophysics and Biomedical Engineering, Bulgarian Academy of Sciences, 1113 Sofia, Bulgaria; 2Biological Faculty, Sofia University “St. Kliment Ohridki”, 8 Dragan Tzankov Str., 1164 Sofia, Bulgaria; 3Clinic of Neurology, Tsaritsa Yoanna University Hospital-ISUL, 1527 Sofia, Bulgaria; 4Department of Anatomy, Histology and Embryology, Medical University-Sofia, Blvd. Sv. Georgi Sofiisky 1, 1431 Sofia, Bulgaria; 5Department of Medical Oncology, University Multi-Profile Hospital for Active Treatment (UMHAT) “St. Ivan Rilski”, 1606 Sofia, Bulgaria; 6Medical College “Y. Filaretova”, Medical University-Sofia, Yordanka Filaretova Str. 3, 1606 Sofia, Bulgaria; 7CSTEP, Office of Continuing Education, Suffolk County Community College 30 Greene Ave., Sayville, NY 11782, USA

**Keywords:** sphingolipid metabolism, ceramide, sphingosine-1-phosphate, quercetin, fingolimod, HepG2 cells

## Abstract

Combinations of anti-cancer drugs can overcome resistance to therapy and provide new more effective treatments. In this work we have analyzed the effect of the polyphenol quercetin and the anti-cancer sphingosine analog fingolimod on the sphingolipid metabolism in HepG2 cells, since sphingolipids are recognized as mediators of cell proliferation and apoptosis in cancer cells. Treatment of hepatocellular carcinoma HepG2 cells with quercetin and fingolimod, alone or in combination, induced different degrees of sphingomyelin (SM) reduction and a corresponding activation of neutral sphingomyelinase (nSMase). Western blot analysis showed that only treatments containing quercetin induced up-regulation of nSMase expression. The same treatment caused elevation of ceramide (CER) levels, whereas the observed alterations in sphingosine (SPH) content were not statistically significant. The two tested drugs induced a reduction of the pro-proliferative sphingolipid, sphingosine 1 phosphate (S1P), in the following order: quercetin, fingolimod, quercetin + fingolimod. The activity of the enzyme responsible for CER hydrolysis, alkaline ceramidase (ALCER) was down-regulated only in the incubations involving quercetin and fingolimod did not affect this activity. The enzyme, maintaining the balance between apoptosis and proliferation, sphingosine kinase 1 (SK1), was down-regulated by incubations in the following order: quercetin, fingolimod, quercetin + fingolimod. Western blot analysis showed down-regulation in SK1 expression upon quercetin but not upon fingolimod treatment. Studies on the effect of quercetin and fingolimod on the two proteins associated with apoptotic events, AKT and Bcl-2, showed that only quercetin, alone or in combination, down-regulated the activity of the two proteins. The reported observations provide information which can be useful in the search of novel anti-tumor approaches, aiming at optimization of the therapeutic effect and maximal preservation of healthy tissues.

## 1. Introduction

Hepatocellular carcinoma (HCC) is one of the most common types of cancer and the third major cause of cancer-related deaths worldwide [1]. This type of cancer is characterized by alterations in various signaling cascades, including the sphingolipid metabolic pathways, resulting in uncontrolled growth of hepatocytes [2]. Polyphenols, which affect largely sphingolipid metabolism, can be combined with classical chemotherapy, to improve the efficacy and decrease the negative side effects of neoplastic drugs. Quercetin (3,3′,4′,5,7-pentahydroxyflavone) is a naturally occurring polyphenolic compound which is distributed in certain fruits and vegetables, especially in red onions (*Allium cepa*), citrus fruits (*Citrus* spp.), red wine, etc. [3]. Due to the presence of phenolic hydroxyl groups, quercetin has been reported to possess a wide range of pharmacological properties, including antioxidant, anti-inflammatory and anti-tumor effects, among others [4,5]. Quercetin can induce cell cycle arrest and apoptosis of HCC cells by modulation of p53, the latter being a tumor suppressor protein involved in cancer prevention, and regulating cell cycle and apoptosis [2,6,7]. Ferrer et al. [8] reported that intravenous administration of quercetin prevented metastatic growth of malignant B16 melanoma F10 cells, by enhancing NO release from the vascular endothelium leading to activation of nSMase and increasing CER and apoptosis. Torres et al. [9] reported that a derivative of quercetin THDF (5,7,30-trihydroxy-3,40-dimethoxyflavone) decreased cell proliferation and induced apoptosis in human leukemia cells by acid SMase-dependent production of ceramide, the latter being correlated with cell death. Another polyphenol, resveratrol, has been reported to significantly alter the levels of the major sphingolipid metabolites and the expression of the enzymes, maintaining their concentration, in HepG2 cells [10].

Sphingolipids are involved in various cellular processes, including cell proliferation, signaling, apoptosis, autophagy, etc. [11]. They have been recognized as mediators of cell death and proliferation in cancer and as potential chemotherapeutics [12]. The most functionally active members of the sphingolipid family are sphingomyelin (SM), ceramide (CER), sphingosine (SPH) and sphingosine-1-phosphate (S1P) [13]. CER is a biologically active sphingolipid that mediates anti-proliferative processes such as apoptosis, cell growth inhibition, senescence modulation, differentiation and autophagy [12]. Its level in cells is regulated via de novo synthesis, hydrolysis through the sphingomyelinase pathway and degradation performed by ceramidases. [14,15]. Enzymes, participating in the SM/S1P pathway play significant roles in cancer onset and progression [16,17]. CER induces cell-type specific apoptosis by activating protein kinase C, protein phosphatases, and by modulating the pro-apoptotic Bcl-2-family proteins [18]. On the contrary, S1P acts as an anti-apoptotic agent by stimulating G protein-coupled receptors activating RAS, RAC and phospholipase C (PLC) [12]. Thus, sphingolipid metabolism occurs as a target for various therapeutic strategies, especially involving cancer cells.

Recently, a great interest was focused on the anti-cancer potential of the synthetic sphingosine analog fingolimod FTY720 (2-amino-2-[2-(4-octylphenyl)]-1,3-propanediolhydrochloride). This is a competitive sphingosine inhibitor of SK1, that has been reported to stimulate relocalization of actin away from the lamellipodia of breast cancer cells, suggesting its possible application for the prevention of tumor metastasis. There is evidence that FTY720 induces apoptosis in prostate and breast cancer cells, and also reduces tumor growth and metastasis [19,20].

In the present work we analyzed the biochemical mechanisms underlying the influence of quercetin and fingolimod (separately and in combination) on the enzymes responsible for the maintenance of the levels of sphingolipids with high physiological activity, which determine the balance between pro-apoptotic and pro-survival processes in hepatocellular carcinoma HepG2 cells. We chose to analyze the effect on the sphingolipid metabolism of these two particular anti-tumor agents, because they are reported to influence sphingolipid metabolites in other cancer cells. For example, quercetin affects SMase in human leukemia cells [9], thus increasing the level of the pro-apoptotic CER, and fingolimod is a structural analog of sphingosine, the latter serving as a precursor of the pro-apoptotic lipid sphingosine-1-phosphate. In addition, in some cases the combined application of drugs shows amplification of their anti-tumor effect, which is why we analyzed the impact of the tested drugs alone or in combination.

## 2. Results

### 2.1. Effect of Quercetin and Fingolimod on Sphingolipid Levels and Neutral Sphingomyelinase

The cytotoxicity of quercetin and fingolimod was determined using different drug concentrations (Figure 1). Cytotoxicity of quercetin was analyzed in a concentration range of 20–120 µM. Cell viability decreased gradually and was about 50% at 80 µM, which was the reason for the choice of this concentration for the experiments. To determine cell viability after incubations with fingolimod cells were treated with the drug in a concentration range of 1–80 µM and incubations in the present study were performed with 20 µM for 24 h (Figure 1B). The viability of the control cells was assumed for 100%. Unlike quercetin, fingolimod did not induce reduction of cell viability at the low doses of the used concentration range.

The influence of quercetin and fingolimod on the SM level in Hep G2 cells is presented in Figure 2. The percentage of SM in the total phospholipids was reduced by 25% as a result of quercetin treatment and remained almost unchanged in fingolimod-treated cells. In cells incubated with a combination of quercetin and fingolimod the observed alteration was quite similar to the one determined in quercetin-treated cells.

Since one major reason for SM reduction could be the activation of neutral sphingomyelinase (nSMase) which hydrolyses SM to CER and phosphocholine, we analyzed this activity in control and treated HepG2 cells (Figure 3). We observed an elevation in nSMase activity as a result of incubation with quercetin, alone or in combination with fingolimod. No statistically significant changes were detected in fingolimod-treated cells. Additionally, Western blot analysis showed that the expression of nSMase in HepG2 cells was increased by 21% only in cells which were treated with quercetin, alone or in combination with fingolimod (Figure 3B).

Besides SM, other biologically active sphingolipid metabolites such as ceramide (CER), sphingosine (SPH) and sphingosine-1-phosphate (S1P) were also altered in Hep G2 cells due to treatment with quercetin and fingolimod (Figure 4). The level of CER was increased most markedly in quercetin-treated cells, followed by the combined treatment. The content of SPH was slightly decreased in all types of treated cells, but the observed differences were not statistically significant. Interestingly, all drugs induced reduction of S1P in the following order: quercetin, fingolimod and quercetin + fingolimod (Figure 4).

### 2.2. Effect of Quercetin and Fingolimod on the Enzymes Maintaining Ceramide Levels

CER is one of the most biologically active sphingolipids and is a product of SMase-induced hydrolysis of SM and/or of de novo biosynthesis. It is closely related to apoptosis initiation, which makes it an important member of the sphingolipid family, especially in cancer cells. CER degradation is performed mainly by alkaline ceramidase (ALCER), an enzyme hydrolyzing CER to SPH, the latter serving as a substrate for a key enzyme in sphingolipid metabolism—SK1. ALCER activity was reduced by 32% due to quercetin treatment (Figure 5). Combined treatment with both tested drugs induced a downregulation of ALCER of 30%. Incubations with fingolimod alone did not affect this enzyme activity (Figure 5A). Western blot analysis did not show any significant differences in ALCER expression between controls and cells treated with the two drugs, alone or in combination (Figure 5B).

### 2.3. Influence of Quercetin and Fingolimod on Sphingosine Kinase 1 Activity and Expression

The two most functionally active sphingolipids, CER and S1P, play opposite roles in determining the fate of cells: CER acts as a pro-apoptotic agent, whereas S1P stimulates cellular proliferation and survival. The key enzyme that controls the balance between these two sphingolipids is SK1, which phosphorylates SPH to produce S1P. Alterations in SK1 regulation produce changes in the level of the pro-proliferative lipid S1P. This is accompanied by corresponding alterations in the balance between apoptosis and survival and these processes are crucial, especially for cancer cells. Our studies showed that quercetin downregulated SK1 in HepG2 cells by 24% (Figure 6A). Western blot analysis revealed that quercetin also down-regulated SK1 expression by about 26% (Figure 6B). In should be noted that treatment with fingolimod induced a stronger down-regulation of SK1 activity (37%), compared to quercetin (Figure 6A). However, it did not affect SK1 expression, as shown in Figure 6B. Finally, the combined treatment with quercetin and fingolimod caused a significant down-regulation of almost 50% of SK1 activity (Figure 6A). The combination of the tested drugs down-regulated SK1 expression by 35%, showing a slight tendency for additional influence of fingolimod on the expression of this protein (Figure 6B).

Studies on the effect of quercetin and fingolimod on the phosphorylation of two proteins associated with apoptotic events, Akt and Bcl-2, showed that quercetin, alone or in combination down-regulated the phosphorylation of Akt and the expression of Bcl-2, whereas fingolimod had no detectable effect (Figure 7).

The suppression of the Akt pathway in cells treated with quercetin alone or in combination with fingolimod suggested that these cells would be more vulnerable to apoptosis. Indeed, when HepG2 cells were subjected to serum starvation for one day, prior to drug incubation, more than 23% of the quercetin-treated cells and about 33% of the cells incubated with both drugs showed apoptotic nuclei, whereas only 5% of the control cells were apoptotic (Figure 8).

## 3. Discussion

Hepatocellular carcinoma is recognized as the third most common cause of cancer-associated deaths in the world [1,21]. Due to its complex etiology the applied therapeutic schemes are various and with differing effectiveness. Thus, there is an increasing need for new anti-cancer approaches, which makes the search for new, more effective and non-toxic drugs an issue of extreme importance. Carcinoma cells alter their metabolism in order to adjust to the malignant phenotype, which results in a decrease in the efficiency of apoptotic events [22]. Several reports imply that chemotherapy-induced apoptosis in many cases is not significant enough to eliminate a high percentage of the cancer cells [23,24]. Application of combined treatments and synergistic induction of apoptosis has been suggested as an appropriate approach to increase the degree of cell elimination in certain types of cancers [25].

As mentioned above, sphingolipids are implicated in essential cellular events and signaling pathways such as proliferation, cancer-genesis, apoptosis, etc. [26,27]. There are numerous studies devoted to the effect of polyphenols on the balance between cell survival and cell death which are of special importance, especially when it comes to cancer cells [28,29,30].

In the present work we focused our attention on the influence of two sphingolipid-related agents, quercetin and fingolimod, on the sphingolipid metabolism in HepG2 cells. Quercetin is a polyphenol with a well-documented pro-apoptotic anti-tumor effect and fingolimod is an inhibitor of SK1, the latter producing the pro-survival sphingolipid S1P. Quercetin induced a rather gradual decrease in cell viability in the used concentration range, whereas fingolimod exhibited a more marked effect at higher concentrations (Figure 1A,B). In our previous studies on the effect of quercetin on the lipid metabolism in three-dimensional tissue-like fibroblast cultures [31] we observed that quercetin affected SM by down-regulation of nSMase, this resulted in an increase in the level of membrane SM. Thus, since quercetin alters sphingolipid metabolism, the latter being implicated in the balance between pro-apoptotic and pro-survival processes, we performed studies on the influence of this polyphenol on the sphingolipids in HepG2 cells. There is evidence that another polyphenol, resveratrol, significantly affects the major sphingolipid metabolites in HepG2 cells [10]. What is more, in our recent paper, we reported that the effect of resveratrol on the lipid metabolism in A549 lung cancer cells is potentiated by combined treatment with fingolimod [32], which is why the latter was included in the present experiments. As evident from Figure 2 treatment of HepG2 cells with quercetin and fingolimod, alone or in combination, induced different effects on the level of SM. Since a major reason for SM decrease could be up-regulation of SMases, we determined the changes in nSMase activity and protein expression (Figure 3A,B). Activation of nSMase was observed only in cells treated with quercetin, alone and in combination, which coincided with the changes found in the SM level, implying that this activity was responsible for the alterations in SM. In addition, Western blot analysis showed that only quercetin up-regulated the expression of nSMase in HepG2 cells (Figure 3B), indicating that this polyphenol affected both the activity and the expression of the main enzyme responsible for CER accumulation, nSMase.

The effect of quercetin, with and without fingolimod, on the major sphingolipid metabolites was also determined (Figure 4). The change of CER levels was statistically significant only in the cases when quercetin was involved in the treatment and fingolimod alone did not induce any noticeable alterations. The level of SPH showed slight changes in all three types of treatments, but the observed differences were not statistically significant compared to controls. However, the level of S1P was gradually decreased in the following order of treatment: quercetin, fingolimod, quercetin + fingolimod, showing that the combined treatment produced a stronger effect on the level of S1P compared to each one of the individual treatments (Figure 4). Accordingly, in our previous studies we observed a decrease in the level of S1P in resveratrol-treated A549 lung adenocarcinoma cells [32]. However, Charytoniuk et al. reported an increase of S1P in resveratrol-treated HepG2 cells in a lipid overload state, which might be related to a more complicated lipid metabolism in cells overloaded with lipid precursors [10]. Nevertheless, these authors reported an increase in active caspase 3, which is indicative of stimulated apoptotic processes.

The enzyme responsible for SPH production from CER, alkaline ceramidase (ALCER), was down-regulated most significantly in cells treated with quercetin, followed by the combined treatment and fingolimod alone (Figure 5). We have reported that resveratrol induced significant changes on the sphingolipid metabolism in A549 cells [32], but it did not affect the activity and expression of ALCER implying a different mechanism of the influence of the two polyphenols, quercetin and resveratrol, on the tested types of cancer cells.

One of the most important sphingolipid-metabolizing enzymes, which determines the capacity of cells to undergo proliferation, is SK1 [33]. Elevated SK1 level is predictive of poor cancer prognosis and overexpression of this enzyme may underlie resistance to chemotherapy [33]. The SK1/S1P signaling pathway has been implicated in the development of various cancers and in resistance to chemotherapeutic drugs [34]. We analyzed the effect of quercetin and fingolimod, alone and in combination, on the activity and protein expression of SK1. Quercetin alone down-regulated SK1 by 24% and the expression showed a reduction of about 26% (Figure 6). It should be noted that treatment with fingolimod induced a stronger down-regulation of SK1 activity compared to quercetin, but its effect on SK1 protein expression was not detectable. Accordingly, our recent studies on the effect of fingolimod on A549 lung adenocarcinoma cells showed a similar effect on SK1—down-regulation of activity and lack of changes in the protein expression [32]. Thus, it seems likely that fingolimod acts in a similar manner in certain types of tumors, which is an important finding concerning the mechanism of action of this sphingolipid analog, since it has also been reported to perform its anticancer effects through both caspase-dependent and caspase-independent pro-apoptotic pathways [33]. Finally, the combined treatment of HepG2 cells with quercetin and fingolimod induced a more significant down-regulation of the activity of SK1 (Figure 6A), implying that the combination of these two agents produces a stronger effect on SK1 activity compared to individual incubations with each one of them. What is more, the combined treatment induced an additional down-regulation in SK1 protein expression, compared to each one of the individual treatments (Figure 8). This is an interesting finding, because the two tested effectors are known as harmless to non-cancer cells, and quercetin is reported to have a beneficial role when applied to normal healthy cells. These results imply that when applied together, quercetin and fingolimod produce a stronger pro-apoptotic effect, compared to cells subjected to individual treatments. Interestingly, fingolimod is currently approved by the Food and Drug Administration for use to treat multiple sclerosis [35], which makes it a drug with a wide range of therapeutic possibilities.

There is evidence that up-regulation of SK1 inhibits doxorubicin-induced apoptosis, associated with the induction of anti-apoptotic proteins such as Bcl-xl, c-IAP1, and TRAF1. Activated Akt phosphorylates and inhibits the pro-apoptotic Bcl-2 family members (e.g., Bad, Bax) and up-regulates the expression of anti-apoptotic Bcl-2 family proteins (e.g., Bcl-2, BclXL) [36]. In the present study, we observed a statistically significant down-regulation in the phosphorylation of Akt and Bcl-2 in cells treated with quercetin, alone and in combination (Figure 7), which was translated to increased apoptosis in HepG2 cells (Figure 8). The observation that fingolimod alone (unlike quercetin) did not alter the phosphorylation of Akt and Bcl-2 expression suggests that the effect of this SPH analog on SK1 was different from the mechanism through which quercetin affected the onset of apoptotic events in HepG2 cells. Thus, in spite of the observed differences, both drugs contributed through different mechanisms to the down-regulation of the major pro-survival enzyme, SK1.

In this work we have focused our attention predominantly on the impact of quercetin and fingolimod on the sphingolipid metabolism, since, to our knowledge, there are no such data in the literature concerning HepG2 cells. The obtained results are a basis for further analysis on apoptotic processes and autophagy induced by agents related to sphingolipid metabolism, which were not analyzed in the present study. It is also of special interest to elucidate whether the observed alterations in the sphingolipid metabolism, induced by quercetin and fingolimod in HepG2 cells, are also valid for other hepatocellular carcinoma cells, which differ in certain signaling pathways and metastatic potential.

In conclusion, these studies demonstrate that the anti-proliferative effect of quercetin on HepG2 hepatocarcinoma cells can be potentiated by its combination with the structural sphingosine analog fingolimod. These two drugs, when applied together, can induce a stronger pro-apoptotic, anti-proliferative effect, which is an important finding, especially when it comes to cancer cells. The reported observations provide information, which can be useful in the search of novel anti-tumor approaches, aiming at optimization of the therapeutic effect with maximal preservation of healthy tissues.

## 4. Materials and Methods

### 4.1. Reagents

Quercetin was purchased from Sigma–Aldrich (St. Louis, MO, USA). DMEM, penicillin–streptomycin (10,000 U·mL^−1^ penicillin and 10,000 mg·mL^−1^ streptomycin) were from Invitrogen (Paisley, UK). DMSO and MTT were from MP Biomedicals, LLC. SPH and S1P were from Avanti Polar Lipids (Alabaster, AL, USA). Quercetin was dissolved in DMSO and was stored at −20 °C until use. Fingolimod was from Cayman Chemicals, Ann Arbor, MI, USA Cat # 11975.

### 4.2. Cell Culture and Incubation with Quercetin and Fingolimod

Hepatocellular carcinoma HepG2 cells were grown in DMEM, 10% FBS (HyClone) (Logan, UT, USA), 100 Units/mL ampicillin and 100 µg/mL streptomycin (Sigma) at 37 °C in a humidified atmosphere, containing 5% CO_2_. The HepG2 cells (5 × 10^5^/well) were seeded in a 24-well plate overnight and treated with of 80 µM quercetin for 24 h. Fingolimod was used at a concentration of 20 µM and incubations were performed for 24 h.

### 4.3. Cell Viability Assay after Incubation with Quercetin and Fingolimod

After the incubations, cell viability was determined by tetrazolium salt measurement (MTT assay), involving assessment of succinate dehydrogenase-induced conversion of 3-[4,5-dimethylthiazol-2-yl]-2,5-diphenyl tetrazolium bromide into formazan crystals. Formation of formazan was measured at 570 nm. The viability of the cells after incubations was estimated as percentage of the absorbance of the treated cells compared to controls.

### 4.4. Determination of Sphingomyelin

This analysis was performed by the Sphingomyelin Quantification Assay Kit (Sigma-Aldrich, Cat.# MAK262). This assay is based on the hydrolysis of sphingomyelin to ceramide and phosphocholine by sphingomyelinase. Alkaline phosphatase (ALP) dephosphorylates phosphocholine to choline producing a reaction that produces a colorimetric signal. After treatment of cells the supernatant was collected, centrifuged to remove cell debris and kept for analysis. The concentration of SM in a sample was calculated as nmol SM per μL.

### 4.5. Determination of Ceramide

The content of ceramide was measured using Ceramide ELISA kit (MyBioSource, Cat.# 3801246) (San Diego, CA, USA) according to the manufacturer’s instructions. The stop solution changed the color from blue to yellow and the intensity of the color was determined at 450 nm. Ceramide concentration was calculated using a standard curve which was generated by plotting the average optical density obtained for each of the standard concentrations.

### 4.6. Determination of Sphingosine

The level of sphingosine was determined using the sphingosine ELISA kit (Aviva Systems Biology, Cat.# OKEH02615) (San Diego, CA, USA). This assay is based on competitive enzyme immunoassay technique. After the final enzymatic reaction, the density of yellow coloration was measured at 450 nm. The obtained value is quantitatively proportional to the amount of biotinylated sphingosine. The standard curve was generated by plotting the optical density at 450 nm vs. the respective standard concentrations.

### 4.7. Determination of Sphingosine-1-Phosphate

Sphingosine-1-phosphate was determined using an ELISA kit for detecting the amount of human sphingosine-1-phosphate in the biological samples (Cat.# 682861, Antibody Research Corporation, St. Charles, MO, USA). This assay is based on the principle of the double-antibody sandwich technique. After treatment of cells, the supernatant was collected, centrifuged and kept for analysis. Then, the procedure described by the manufacturer was followed and the optical density of the samples was measured at 450 nm. The amount of S1P in the samples was determined using a standard curve.

### 4.8. Neutral Sphingomyelinase Activity Assay

Neutral sphingomyelinase activity was measured by the Neutral Sphingomyelinase Activity assay Kit (Echelon Biosciences, Cat.# K-1800) (Salt Lake City, UT, USA). In this enzyme-coupled assay, nSMase catalyzes the hydrolysis of sphingomyelin to ceramide and phosphorylcholine. Alkaline phosphatase then catalyzes the dephosphorylation of phosphorylcholine to choline. Choline is oxidized by choline oxidase to form hydrogen peroxide. In the presence of 4-Aminoantipyrine (4 AAP), hydrogen peroxide and peroxidase oxidize 4 AAP to form a blue chromogen, which was determined by measuring the absorbance at 595 nm.

### 4.9. Alkaline Ceramidase Activity Assay

Alkaline ceramidase activity was determined by measuring the released sphingosine from ceramide [37]. Briefly, the cells (75 µg protein) were incubated with ceramide (200 µM) in 25 mM Tris–HCl buffer, pH 9, containing 5 mM CaCl_2_ and 5 mM MgCl_2_ at 37 °C for 60 min. The reaction was stopped by addition of 0.5 mL CHCl_3_/CH_3_OH (2:1, *v*/*v*). The produced sphingosine was quantified by a sphingosine ELISA kit (Aviva Systems Biology, Cat.# OKEH02615).

### 4.10. Sphingosine Kinase 1 Activity Assay

Sphingosine kinase 1 activity was determined using a sphingosine kinase 1 assay ELISA kit (SPK1 DuoSet ELISA, R&D Systems, Cat.# DY5536) (Minneapolis, MN, USA) as recommended by the manufacturer’s instructions. The optical density was determined after addition of the stop solution at 450 nm.

### 4.11. Western Blotting and Antibodies

Control and treated cells were solubilized in RIPA buffer (150 mM NaCl, 2 mM EDTA, 1% sodium deoxycholate, 0.1% SDS, 1% Triton X-100, 10% glycerol, 50 mM HEPES, pH 7.5) containing Complete Inhibitory Cocktail (Cat.#11697498001, Merck, Rahway, NJ, USA). Equal volumes of 5× sample buffer (60 mM Tris-HCl, pH 6.8, 2% SDS, 10% glycerol, 5% b-mercaptoethanol and bromophenol blue) were added and the samples were heated for 4 min at 95 °C. The three replicate samples from control and treated cells from each individual experiment were combined and ran on a single lane. Proteins were resolved on SDS-PAGE, transferred toa nitrocellulose membrane and blocked for 1 h in 5% non-fat dry milk in TBST (50mM Tris base, 200 mM NaCl, 0.1% Tween-20, pH 7.4). Membranes were than incubated with the appropriate primary and secondary antibodies, including monoclonal anti-neutral sphingomyelinase 2 (Cat.# sc-166637), anti-sphingosine kinase 1 (Cat.# sc-365401), anti-GAPDH (Cat.# sc-32233), anti-phospho-Akt (sc-293125), anti-total Akt (sc-271149), anti-Bcl-2 (sc-7382) all from Santa Cruz Biotechnology (Dallas, TX, USA), anti-alkaline ceramidase 2 (Cat.# A09706-1, Boster Biological Technology), anti-mouse IgG (Fab) HRP conjugate (Cat.# SAB-100, Stressgen), anti-rabbit IgG (Fab) HRP conjugate (Cat.# SAB-300, Stressgen). Immunoblots were visualized using the ECL system (Santa Cruz).

### 4.12. Apoptosis Assay

Apoptosis induced by quercetin and fingolimod was tested by serum starvation of HepG2 cells for 24 h and incubation with the drugs for another 24 h, again in the absence of serum. After these treatments, the cells were fixed with 4% paraformaldehyde in phosphate-buffered saline containing 5% sucrose for 20 min and stained without permeabilization with 1 µM Hoechst 33,342 fluorochrome (PureBlu™ Hoechst 33,342 Nuclear Staining Dye, Bio-Rad, Hercules, CA, USA) for 5 min. Immunofluorescent images were obtained with a DeltaVision Ultra™ (GE Healthcare, Chicago, IL, USA) microscope. Digital images, ImageJ Image Processing and Analysis software (Version 1.53a, NIH, Bethesda, MD, USA) were used to score the number of apoptotic (bright) and non-apoptotic (dark) nuclei from two randomly chosen fields from each sample obtained from three separate experiments.

### 4.13. Statistical Analysis

Statistical processing of the data was made by one-way analysis of variance (ANOVA) using In Stat software Graph Pad In Stat 3.1, developed by Graph Pad Software, San Diego, CA, USA. Data analyzed were obtained from three independent experiments, each containing three replicates for the controls and three replicates for the respective treatments.

## 5. Conclusions

Quercetin, but not fingolimod, down-regulates neutral sphingomyelinase and induces reduction of sphingomyelin in hepatocellular carcinoma HepG2 cells.Quercetin reduces the content of ceramide, whereas the combined treatment of quercetin with fingolimod most significantly affects the level of sphingosine-1-phosphate.Quercetin treatment down-regulates the activity, but not protein expression, of alkaline ceramidase in HepG2 cells.Combined treatment of HepG2 cells with quercetin and fingolimod induces a stronger down-regulation of sphingosine kinase 1, compared to individual treatments with each one of them.Treatment with quercetin, but not with fingolimod, reduced the phosphorylation of Akt and Bcl-2 expression, leading to increased apoptosis in HepG2 cells.

## Figures and Tables

**Figure 1 ijms-23-13916-f001:**
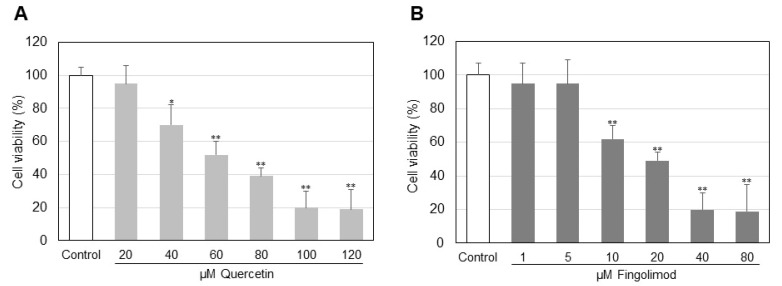
Viability of HepG2 cells treated for 24 h with quercetin in a concentration range of 20–120 μM (**A**) and fingolimod (**B**) in a concentration range of 1–80 μM. Statistical significance is presented as comparison of treated cells with controls. * *p*< 0.05; ** *p*< 0.001.

**Figure 2 ijms-23-13916-f002:**
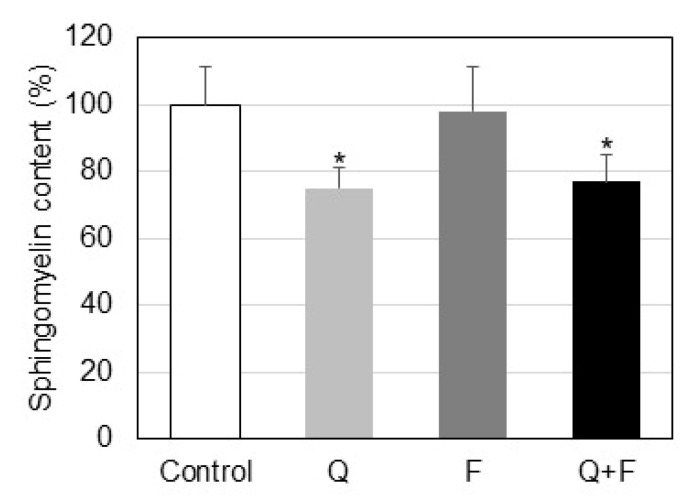
Sphingomyelin content in HepG2 cells untreated (Control) and treated with quercetin (Q) and fingolimod (F). Values are expressed as relative percentage participation in the total lipids. Values are means ± SD. Statistical significance is presented as comparison of treated cells with controls. * *p* < 0.01.

**Figure 3 ijms-23-13916-f003:**
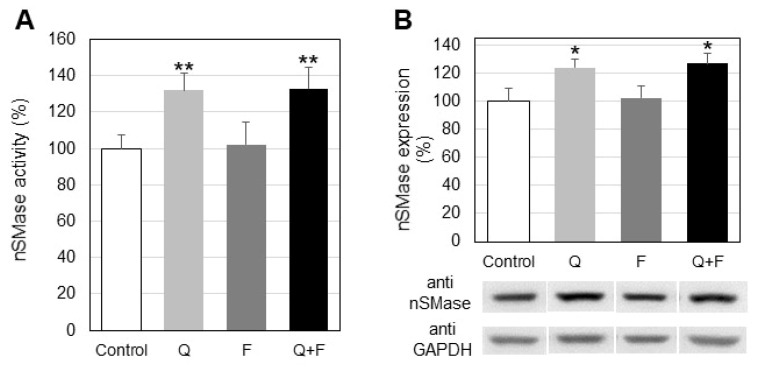
Changes in the activity (**A**) and protein expression (**B**) of neutral sphingomyelinase (nSMase) in control and treated with quercetin (Q) and fingolimod (F) HepG2 cells. Values are expressed as % of controls (100%). Representative images from Western blot analysis with specific antibodies to neutral sphingomyelinase (anti-nSMase) are shown at the lowermost of panel B. Reaction with anti-glyceraldehyde-3-phosphate dehydrogenase antibodies (anti-GAPDH) was used as an internal control for loading. Values are means ± SD. Statistical significance is presented as comparison of treated cells with controls. * *p* < 0.05; ** *p* < 0.01.

**Figure 4 ijms-23-13916-f004:**
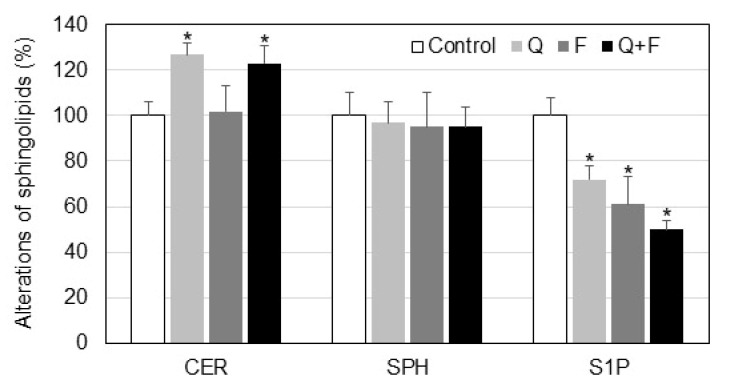
Changes in the level of ceramide (CER), sphingosine (SPH) and sphingosine-1-phosphate (S1P) in control and treated with quercetin (Q) and fingolimod (F) HepG2 cells. Values are expressed as % of controls. Values are means ± SD. Statistical significance is presented as comparison of treated cells with controls. * *p* < 0.01.

**Figure 5 ijms-23-13916-f005:**
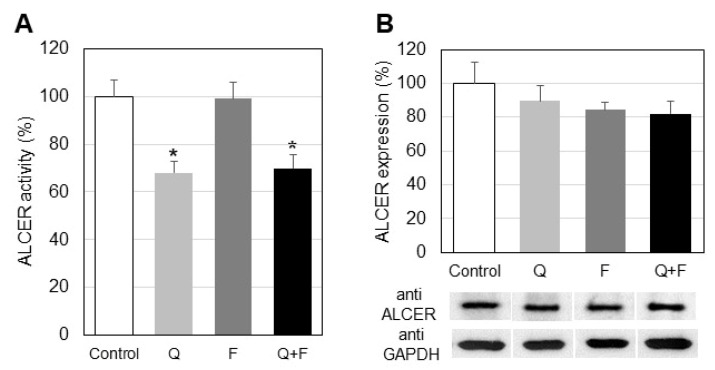
Alteration in the activity (**A**) and protein expression (**B**) of alkaline ceramidase (ALCER) in control and treated with quercetin (Q) and fingolimod (F) HepG2 cells. Values are expressed as % of controls (100%). Representative images from Western blot analysis with specific antibodies to alkaline ceramidase (anti-ALCER) are shown at the bottom of panel B. Reaction with anti-glyceraldehyde-3-phosphate dehydrogenase antibodies (anti-GAPDH) was used as an internal control for loading. Values are means ± SD. Statistical significance is presented as comparison of treated cells with controls. * *p* < 0.01.

**Figure 6 ijms-23-13916-f006:**
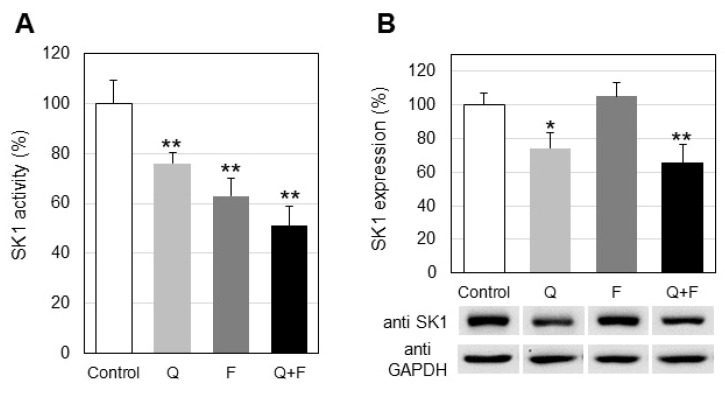
Changes in the activity (**A**) and protein expression (**B**) of sphingosine kinase 1 (SK1) in control and treated with quercetin (Q) and fingolimod (F) HepG2 cells. Values are expressed as % of controls (100%). Representative images from Western blot analysis with specific antibodies to sphingosine kinase 1 (anti-SK1) are shown at the lowermost of panel B. Reaction with anti-glyceraldehyde-3-phosphate dehydrogenase antibodies (anti-GAPDH) was used as an internal control for loading. Values are means ± SD. Statistical significance is presented as comparison of treated cells with controls. * *p* < 0.05; ** *p* < 0.01.

**Figure 7 ijms-23-13916-f007:**
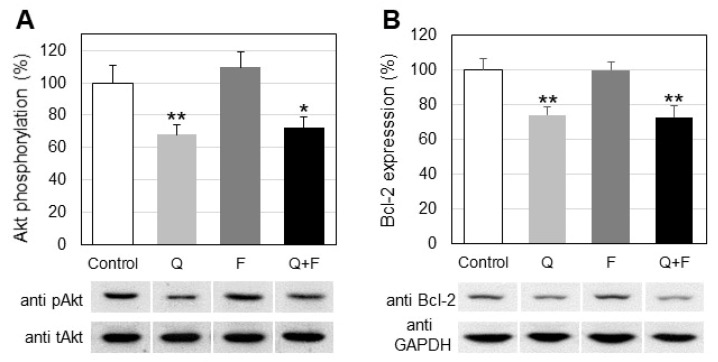
Changes in Akt phosphorylation (**A**) and Bcl-2 expression (**B**) in control and treated with quercetin (Q) and fingolimod (F) HepG2 cells. Values are expressed as % of controls. Representative images from Western blot analysis with specific antibodies to phosphorylated Akt (anti-pAkt) and B cell lymphoma 2 protein (anti-Bcl-2) are shown at the lowermost of the panels. Reactions with antibodies against total Akt (tAkt) and anti-glyceraldehyde-3-phosphate dehydrogenase antibodies (anti-GAPDH) were used as internal controls for loading. Values are means ± SD. Statistical significance is presented as comparison of treated cells with controls. * *p* < 0.05; ** *p* < 0.01.

**Figure 8 ijms-23-13916-f008:**
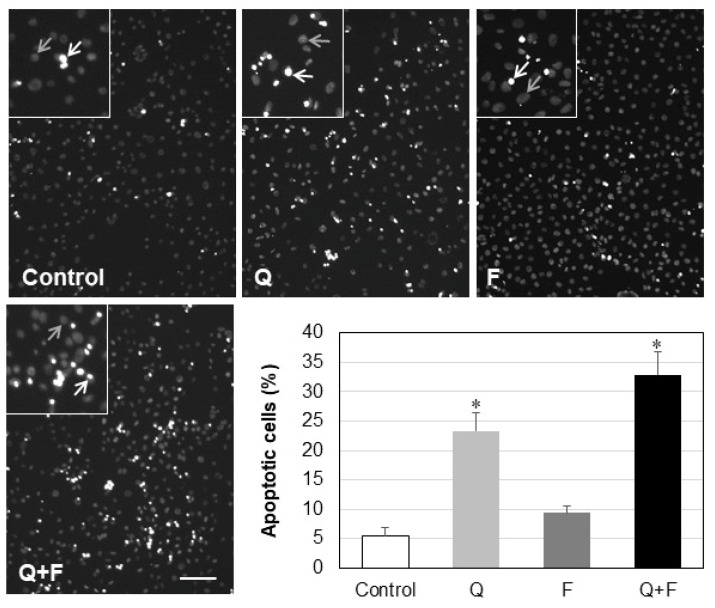
Apoptotic response of HepG2 cells after treatment with quercetin (Q), fingolimod (F) or combination of both drugs (Q + F). Apoptotic (white arrows in the insets) and non-apoptotic (gray arrows in the insets) nuclei were scored, and the percentage of apoptotic cells in each sample was determined. Bar, 100 µm. * *p* < 0.001.

## Data Availability

Data is contained within the article.

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
