# Peer review of "Effect of Quercetin and Fingolimod, Alone or in Combination, on the Sphingolipid Metabolism in HepG2 Cells"

_ijms, 2022, doi:10.3390/ijms232213916_

Round 1

Reviewer 1 Report

 Paper ID

 Title: Effect of quercetin and fingolimod, alone and in combination, on the sphingolipid metabolism in HepG2 cells

Authored by: Albena Momchilova, Georgi Nikolaev, Stefan Pankov, Evgenia Vassileva, Nikolai Krastev, Bozhil Robev, Dimo Krastev, Adriana Pinkas and Roumen Pankov

The manuscript by Momchilova and co-workers presents data from multiple techniques on the molecular mechanisms of quercetin and fingolimod as well as their combination on the sphingolipid metabolism in HepG2 cells.

The data are of high relevance to not just the field of biochemistry and cancer research but have a broader appeal to numerous biomedical studies that have addressed the positive pleiotropic effect of natural compounds on health and diseases.

The data suggest that the anti-proliferative effect of quercetin on HepG2 hepatocarcinoma cells can be potentiated by its combination with the structural sphingosine analogue fingolimod.

 The strengths of the manuscript:

1.      The use of several appropriate methods and approaches: cell viability  assay, sphingomyelin, ceramide, sphingosine and sphingosine-1-phosphate quantification assays, neutral sphingomyelinase, alkaline ceramidase and sphingosine kinase 1 activity assays, apoptosis assay and Western blotting.

2.      The results are presented in a very clear and precise way that makes the manuscript easily readable and intriguing to the reader. The authors give new original data related to the search for novel anti-tumor approaches aiming at optimization of the therapeutic effect with maximal preservation of healthy tissues.

3.     The authors correctly make the interpretation of the obtained results and their discussion in light of the previously published data on this subject.

4.  The conclusions are exact and correspond to the obtained results.

5.      Enough recent references are used to justify and support the presented study.

Based on the above-mentioned arguments, I consider that the manuscript is ready to be published as it is.

Author Response

The strengths of the manuscript:

Reviewer: The use of several appropriate methods and approaches: cell viability  assay, sphingomyelin, ceramide, sphingosine and sphingosine-1-phosphate quantification assays, neutral sphingomyelinase, alkaline ceramidase and sphingosine kinase 1 activity assays, apoptosis assay and Western blotting.

Answer: We would like to thank this Reviewer for the high estimation of our methodology.

Reviewer: The results are presented in a very clear and precise way that makes the manuscript easily readable and intriguing to the reader. The authors give new original data related to the search for novel anti-tumor approaches aiming at optimization of the therapeutic effect with maximal preservation of healthy tissues.

Answer: Again, we would like to thank this Reviewer for finding this paper readable and intriguing to the readers.

Reviewer: The authors correctly make the interpretation of the obtained results and their discussion in light of the previously published data on this subject.

Answer: We are thankful for these positive words.

Reviewer: The conclusions are exact and correspond to the obtained results.

Answer: Again, we thank this Reviewer for finding the conclusions drawn in this work corresponding to the reported results.

Reviewer: Enough recent references are used to justify and support the presented study.

Answer: Thanks again for the support of this Reviewer.

Reviewer 2 Report

This study was aimed to evaluate the “effect of quercetin and fingolimod, alone and in combination, on the sphingolipid metabolism in HepG2 cells”.

My concerns and comment of this study are listed below.

1. The rationale of use of quercetin and fingolimod on the metabolism of sphingolipid was not described in the “introduction”. Authors please explain how them decide to use quercetin and fingolimod, alone and in combination, on the sphingolipid metabolism in HepG2 cells.

2. Authors please provide more details of biomarkers of the extrinsic or intrinsic pathway of apoptosis of hepatoma cell lines affected by the impact of quercetin and fingolimod on the metabolism of sphingolipid. The current data is insufficient to discuss the roles of quercetin and fingolimod on apoptosis of HepG2 cells.   

3.There was no data of the effect of autophagy of hepatoma cell lines affected by the impact of quercetin and fingolimod on the metabolism of sphingolipid in this study. Authors please provide these laboratory data.

4.Authors may use multiple hepatoma cell lines to verified their hypothesis investigated in this study.

Author Response

Reviewer: The rationale of use of quercetin and fingolimod on the metabolism of sphingolipid was not described in the “introduction”. Authors please explain how them decide to use quercetin and fingolimod, alone and in combination, on the sphingolipid metabolism in HepG2 cells.

Answer: We thank this reviewer for his/her recommendation. We agree that the use of the two effectors should be justified by more arguments, which was done in the amended version by adding to the end of the “Introduction” the following text: “We chose to analyze the effect on the sphingolipid metabolism of these two particular anti-tumor agents, because they are reported to influence sphingolipid metabolites in other cancer cells. For example quercetin affects SMase in human leukemia cells [9], thus increasing the level of the pro-apoptotic ceramide, and fingolimod is a structural analog of sphingosine, the latter serving as precursor of the pro-apoptotic lipid sphingosine 1 phosphate. In addition, in some cases the combined application of drugs shows amplification of their anti-tumor effect, which is why we analyzed the impact of the tested drugs alone and in combination.”

Reviewer: Authors please provide more details of biomarkers of the extrinsic or intrinsic pathway of apoptosis of hepatoma cell lines affected by the impact of quercetin and fingolimod on the metabolism of sphingolipid. The current data is insufficient to discuss the roles of quercetin and fingolimod on apoptosis of HepG2 cells.  

Answer: This paper is devoted to analysis of the effect of quercetin and fingolimod on the sphingolipid metabolism in HepG2 cells. Our aim was not to assess the occurrence of apoptotic processes, but to follow the changes in the sphingolipid pathway, induced by the two effectors, alone and in combination, and to reveal the biochemical basis, underlying the observed changes in the levels of biologically active sphingolipids. Nevertheless, we have analyzed some of the intrinsic markers, by which the altered sphingolipids could induce onset of apoptotic processes, such as Akt phosphorylation and Bcl-2 expression, which implied that quercetin-treatment made HepG2 cells more vulnerable to apoptosis, compared to untreated cells. In addition, we presented data for the physiological outcome – the increased percentage of apoptotic nuclei in treated cultures of HepG2 (fig.8). However, I would like to emphasize again, that the main goal of this work was to follow the changes in the sphingolipid metabolites and the enzymes responsible for their alterations, as this is clearly stated in the title.

Reviewer: There was no data of the effect of autophagy of hepatoma cell lines affected by the impact of quercetin and fingolimod on the metabolism of sphingolipid in this study. Authors please provide these laboratory data.

Answer: Autophagy is a very interesting issue, which deserves special attention, but it was not analyzed in the present work.

Reviewer: Authors may use multiple hepatoma cell lines to verified their hypothesis investigated in this study.

Answer: We thank the reviewer for this relevant suggestion. In fact, we are investigating the effect of quercetin and fingolimod on Hep3B and the results show changes in sphingolipid metabolism in the same direction as HepG2, but with some interesting deviations. The obtained results can be explained by the significant differences that exist between these two cell lines (Qiu, G. et al., (2015). Distinctive pharmacological differences between liver cancer cell lines HepG2 and Hep3B. Cytotechnology, 67(1), 1-12). These intriguing changes are the subject of a separate scientific publication that we are preparing.

Round 2

Reviewer 2 Report

Authors  had adequate response to my comment 1.

However, authors please  modify your replies to my comment 2,3, and 4 and integrate these modified replies as the limitations of this current study into the part of discussion.

For example, authors emphasize the importance of autophagy related to  sphingolipids  in the part of introduction. Thus,authors please explain why didn't authors analyze the expression of  autopahgy of hepatoma cell lines  in this study?  Therefore, readers can accept the laboratory data presented in this manuscript can support the conclusions claimed by authors.

Author Response

Reviewer 2: Authors had adequate response to my comment 1.

However, authors please  modify your replies to my comment 2,3, and 4 and integrate these modified replies as the limitations of this current study into the part of discussion.

For example, authors emphasize the importance of autophagy related to  sphingolipids  in the part of introduction. Thus, authors please explain why didn't authors analyze the expression of  autopahgy of hepatoma cell lines  in this study?  Therefore, readers can accept the laboratory data presented in this manuscript can support the conclusions claimed by authors.

Answer: Based on the reviewer’s recommendation, we have included the following text, pointing out the limitations of the current study, at the end of the Discussion: “In this work we have focused our attention predominantly on the impact of quercetin and fingolimod on the sphingolipid metabolism, since, to our knowledge, there are no such data in the literature concerning HepG2 cells. The obtained results are a basis for further analysis on apoptotic processes and autophagy induced by agents related to sphingolipid metabolism, which were not analyzed in the present study. It is also of special interest to elucidate whether the observed alterations in the sphingolipid metabolism, induced by quercetin and fingolimod in HepG2 cells, are also valid for other hepatocellular carcinoma cells, which differ in certain signaling pathways and metastatic potential.”

Also, we would like to point out that we have not emphasized in the Introduction the importance of autophagy as related to sphingolipid metabolism, but we have cited that sphingolipids affect processes like differentiation, senescence and autophagy, among others. None of these processes was a subject of the present studies.

Again, we would like to thank this Reviewer for his/her valuable recommendations and suggestions.